# Structural Properties of Interfacial Layers in Tantalum to Stainless Steel Clad with Copper Interlayer Produced by Explosive Welding

**Henryk Paul \*** , **Robert Chulist and Izabela Mania**

Institute of Metallurgy and Materials Science, Polish Academy of Sciences, 25 Reymonta St., 30-059 Krakow, Poland; r.chulist@imim.pl (R.C.); i.mania@imim.pl (I.M.)
\* Correspondence: h.paul@imim.pl; Tel.: +48-12-2952833

**Abstract:** A systematic study of explosively welded tantalum and 304 L stainless steel clad with M1E copper interlayer was carried out to characterize the microstructure and mechanical properties of interfacial layers. Microstructures were examined using transmission and scanning (SEM) electron microscopy, whereas mechanical properties were evaluated using microhardness measurements and a bending test. The macroscale analyses showed that both interfaces between joined sheets were deformed to a wave-shape with solidified melt zones located preferentially at the crest of the wave and in the wave vortexes. The microscopic analyses showed that the solidified melt zones are composed of nano-/micro-crystalline phases of different chemical composition, incorporating elements from the joined sheets. SEM/electron backscattered diffraction (EBSD) measurements revealed the microstructure of layers of parent sheets that undergo severe plastic deformation causing refinement of the initial grains. It has been established that severely deformed areas can undergo recovery and recrystallization already during clad processing. This leads to the formation of new stress-free grains. The microhardness of welded sheets increases significantly as the joining interface is approaching excluding the volumes directly adhering to large melted zones, where a noticeable drop of microhardness, due to recrystallization, is observed. On lateral bending the integrity of the all clad components is conserved.

**Keywords:** explosive welding; tantalum/copper/stainless steel clads; severe plastic deformation; SEM/EBSD; microhardness

## 1. Introduction

New strategies in the development of metallic materials for advanced structural applications involve the synthesis of bulk compounds that contain metallurgical bond. The bi- or multi-layered composites with built-in specific functionalities are an example of such materials. They offer an optimum balance between manufacturing and service costs, and the durability to perform under various conditions of usage. For materials used in the chemical industry, the proper combination of strength and high anticorrosive resistance, usually at high temperatures are especially important. The tantalum (Ta) and stainless steel composite is one of the industrially relevant bi-layered metallic material used in this field [1–4]; tantalum provides excellent corrosion resistance, while the stainless steel substrate is typically used as a load-bearing component. In most of the corrosion situations, it is capable to protect all installations exposed to highly oxidizing or caustic environments, where glass-lined equipment is subject to mechanical damage or thermal shock failures. Since the high cost of tantalum has traditionally been a major impediment in wide-scale industrial applications, such as large pressure vessels, therefore, it is advisable to use them rather as a coating on carbon, stainless or

'duplex'-type steels. Ta cladding is often used as an alternative to Ta coatings for fabricating coating parts out of solid Ta. Compound materials of this type (e.g., large vessels/tanks) are both structurally and cost-effective.

The Ta/stainless steel composites in the form of sheets/plates are difficult (or impossible) to produce via conventional methods of joining due to metallurgical incompatibility between joined components (high difference in the melting points of these metals, Ta at 3290 K and Fe at 1811 K). Therefore, explosive welding (EXW) is, at present, the only efficient way of surface joining of Ta and stainless steel sheets (Figure 1). However, further processing of bi-layered Ta/stainless or carbon steel composites is strongly restricted. This results from serious difficulties of a butt joint formation during 'conventional' welding due to limited heat transfer from the heat-affected zone. Moreover, if one attempts to use fusion welding to joint Ta to steel, the molten pool tends toward the eutectic composition (they are formed even well below the melting point of Ta) and then form brittle intermetallics upon solidification [1]. To solve problems associated with conventional welding of the Ta/stainless steel clads an intermediate layer, made of soft material and high thermal conductivity, such as copper (Cu), is used. Besides rapid heat dissipation, Cu has one more advantage, i.e., it does not react metallurgically with Ta.

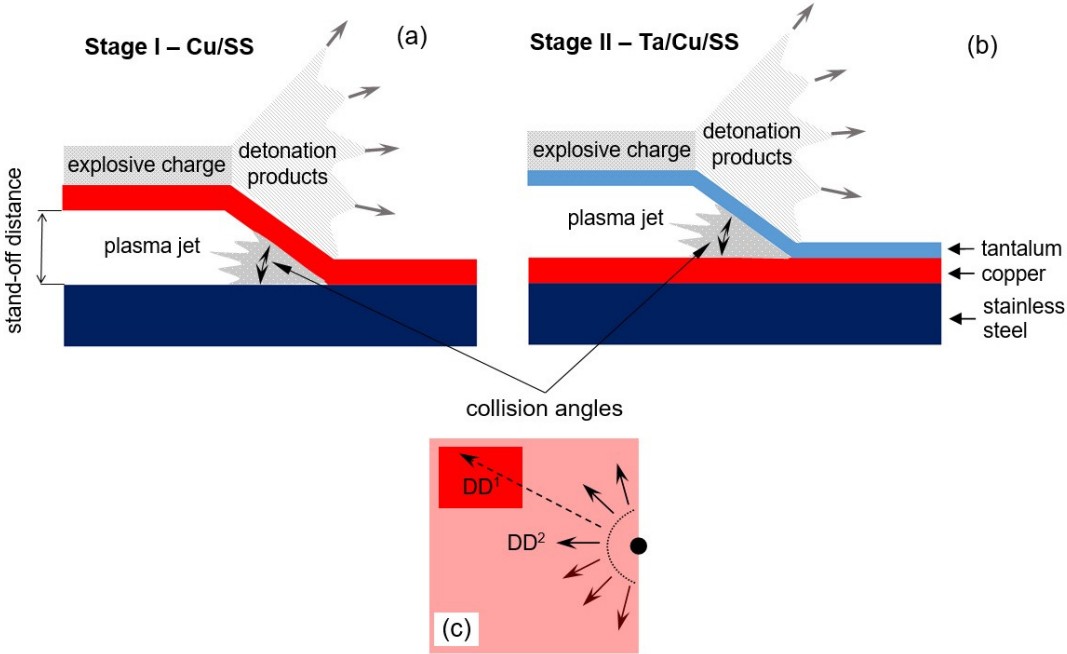

**Figure 1.** (**a**) Schematic representation of explosive welding in two steps: (**a**) stage I—formation of a Cu/stainless steel (SS) clad, (**b**) stage II—formation of Ta/Cu/SS clad, and (**c**) cutting of Cu/SS base plate before the second stage of joining.

In earlier works on different metal compositions, a lot of attention was put into an explanation of the correlation between the clad strength and the interface waviness and the quantity of solidified melt zones, e.g., [5,6]. However, it becomes increasingly apparent that not the details of the interface waviness but the complex microstructure of interfacial layers determines the mechanical and some physical properties of the clad [7–13]. In the light of such evidence, various microstructural transformations can be distinguished. On the one hand, the shear stresses, which occur due to oblique collision of the sheets are responsible for strain hardening and turbulent flow of the interfacial layers. This leads to the formation of wavy interfaces between the joined sheets. Since the interfacial layers are subjected to severe plastic deformation [14–18] they can easily undergo recovery and recrystallization. On the other hand, the processes of fast heating followed by fast cooling during clad preparation result in the formation of solidified melt zones of different structures, phase composition and mechanical properties [7–11,19,20].

Explosive welding of Ta and other metals have received much attention so far, referring to, e.g., Ta/Cu/steel [5,21], Cu/steel [22–24] and Cu/Ta [25–27]. The Cu–Ta system is characterized by nearly zero mutual solubility of the components in the solid-state [28,29] and high structural and mechanical stability at elevated temperatures. Greenberg et al. [30], Maliutina et al. [31] as well as Bataev et al. [11] have shown that explosively welded Ta and Cu sheets exhibit a heterophase mixture in the reaction region with the size of the dispersed Ta and Cu particles similar to those of colloids. As also documented by Bataev et al. [11] and Parchuri et al. [32], the $Ta_xCu_{1-x}$ based intermetallics or decagonal quasicrystals were found to coexist along with pure Ta and Cu particles in the solidified melt zones at the Ta/Cu interface. The formation of metastable phases can also be expected due to rapid cooling during the solidification of melted volumes. In earlier works, the metastable phases in Ta–Cu system were investigated by Cullis et al. [33] who observed the metastable substitutional solid solution in the form of thin films. Furthermore, amorphous phases were observed by Natasi et al. [34] and Gong et al. [35], whereas the nano-crystalline phases by Purja Pun et al. [36] and Rajagopalan et al. [37]. On the other hand, a Cu/stainless steel interface contains solidified melt zones that are exclusively composed of intermetallic phases of different chemical compositions and various morphology of grains, e.g., [16,19]. Moreover, independently on metals composition, radical temperature changes [10,11] can lead to remarkable microstructural transformation in near-the-interface layers of the bonded sheets. In earlier works, a lot of attention was paid to the role of solidified melt zones with respect to strength properties, whilst significantly less effort has been directed towards characterizing their internal microstructure. To the best of the authors' knowledge, there is an absence of detailed studies of the Ta/Cu/stainless steel metals combination that specifically discussed the strain hardening, recovery, and recrystallization of interfacial layers.

Therefore, this work is intended to show interconnected phenomena that must be considered in the interfacial layers of joined sheets in the Ta/Cu (M1E)/(304L stainless steel) composite at the Ta/Cu (M1E) and Cu/304 L stainless steel interfaces. The microstructures and chemical composition changes are analyzed using scanning (SEM) and transmission (TEM) electron microscopes equipped with energy dispersive spectrometry (EDS) detectors. Since the interfacial layers are subjected to severe plastic deformation, which can undergo partial recrystallization, the high-resolution electron backscattered diffraction (EBSD) facility was used as a suitable tool to study the microstructural changes occurring in the parent sheets (areas of not mixed original sheets excluding melted zones). In order to support microstructural findings, the mechanical properties were evaluated using microhardness measurements, whereas the integrity of the joints via lateral bending test.

## 2. Experiment

EXW of Ta/Cu/stainless steel (SS) sheets was performed in two steps by High Energy Technologies Works 'Explomet' (Opole, Poland). In the first step (Figure 1a) the explosive welding of M1E Cu (flyer) to 304 L SS—base sheets with a size of 2400 mm × 2400 mm—was performed. The chemical composition of the joined sheets are presented in Table 1. After straitening the new plate was cut from the corner region of the Cu/SS plate (but still within the area of properly bonded sheets), as presented in Figure 1b. The dimension of this new, bimetallic sheet was 440 mm (length) × 205 mm (width). Then, in the second stage (Figure 1c), the Ta (flyer) sheet was clad onto the Cu/SS (base) plate. It is clear that the detonation direction during first (DD[1]) and the second (DD[2]) EXW steps are not parallel; the DD[2] is inclined at ~35° with respect to DD[1]. The initial thicknesses of the sheets were: 1.8 mm (Ta), 3.0 mm (Cu) and 12 mm (SS). The contact surfaces of the joined sheets/plates were grounded, cleaned of solid particles and degreased. A detonator was located in the middle of the shorter edge of the flyer plate. The explosive was ammonium nitrate with fuel oil and a charge density of amount 0.75 $g/cm^3$. To manufacture high-quality clads, the detonation velocities during both steps of the EXW experiments were ranged between 2500–2600 m·s$^{-1}$. The welding conditions were tailored through the parallel geometry route with a 3 mm stand-off distance between the sheets (on each step).

**Table 1.** Chemical composition of joined components.

| **304 L Steel (Arcelor Mittal Certificate)** | | | | | | | | | | |
|---|---|---|---|---|---|---|---|---|---|---|
| Chemical element | C | Mn | P | S | Si | Cu | Ni | Cr | Mo | Co | Fe |
| wt. % | 0.24–0.30 | 1.87–2.0 | 0.028–0.045 | 0.0017–0.015 | 0.323–0.75 | 0.257–0.750 | 8.037–10,5 | 18.035–19.5 | 0.238–0.75 | 0.129 | balance |

| **Tantalum (Hamilton Precision Metals®Certificate)** | | | | | | | | | |
|---|---|---|---|---|---|---|---|---|---|
| Chemical element | C | O | N | H | Ni | Ti | W | Mo | Si | Ta |
| wt. % | 0.01 | 0.015 | 0.01 | 0.0015 | 0.1 | 0.1 | 0.05 | 0.02 | 0.005 | balance |

| **M1E—Copper (Carl Schreiber GmbH Certificate)** | | | | | | | | | |
|---|---|---|---|---|---|---|---|---|---|
| Chemical element | | | | | ppm | | | | | wt.% |
| | Ag | Ni | Fe | Sb | As | Sn | Zn | S | O | Cu |
| ppm/wt.% | 12.0 | 3.0 | 2.0 | 2.0 | 1.7 | 1.7 | 1.7 | 5.0 | 30.0 | 99.95 |

Specimens for microstructural analyses were cut-off from the central part of the final clad in the as-welded state. The observation plane was perpendicular to the transverse direction (TD). This means that sample edges were parallel to the detonation ($DD^2$) and to the normal (ND) directions. The samples were mechanically ground up to 4000 SiC paper and polished in two steps with the use of the VibroMet-2 (Buehler, Lake Bluff, IL, USA) device and $Al_2O_3$ for 10 h and colloidal silica for 2 h. To study the microstructure evolution a high-resolution SEM (FEI Quanta 3D, Tokyo, Japan) equipped with EDS detector and high-speed Hikari EBSD camera by EDAX, were used. During SEM/EBSD measurements, the microscope control, pattern acquisition, and indexing were done using the Genesis TSL OIM Analysis 8 software (EDAX, Weiterstadt, Germany). The mappings were carried out in the beam-scanning mode. The applied step size ranged between 40 nm and 200 nm and with an accelerating voltage ranging between 15 and 30 kV. Supplementary analyses on the nanoscale were performed using TEM, FEI Technai Super Twin $G^2$ FEG (Tokyo, Japan) operating at 200 kV, equipped with an energy dispersive X-ray microanalysis system.

Vickers microhardness measurements were performed on the $ND/DD^2$ section to estimate the microhardness of intermetallic phases and the distribution in strain hardened layers across the interface. The tests were carried out on a finely-polished longitudinal section. The obtained microhardness values were the average of three indentation measurements. The average microhardness of the base materials in fully recrystallized states, i.e., before cladding, were 250 HV, 162 HV, and 110 HV for 304 L stainless steel, Ta and M1E-Cu, respectively. A three-point lateral bending test was employed to evaluate the resistance to delamination of the joints. The test was performed according to EN 13445-2:2014 (E) on samples cut along the $DD^2$ from the final clad. The specimens for bending testing with a size of 10 mm (width) × 13 mm (high) × 150 mm (length) were extracted from the central part of the clad in the plane parallel to the detonation direction.

## 3. Results

### 3.1. Macro-/Meso-Scale Interfaces Overview

Macroscale characterization of the interfacial layers includes light optical microscopy and low magnification SEM observations. The initial state of the sheets was characterized by a uniform, fully recrystallized microstructure. Both sections perpendicular to the rolling plane revealed structures of equiaxed grains with a diameter of ~80 μm, ~60 μm and ~100 μm for Ta, 304L (SS) and Cu, respectively (Figure 2). It confirmed the high quality of the Ta/Cu and Cu/SS interfaces, without voids and visible sheets delamination. The analysis made at the mesoscopic scale with low magnification SEM imaging shows wavy interfaces, however, with quite different wave parameters (Figure 3). Moreover, the character of waviness is different for both interfaces since the detonation direction during the first ($DD^1$) and the second ($DD^2$) EXW steps are not parallel. In the case of Ta/Cu interface the amplitude and the period of the wave, as observed in the $ND/DD^2$ section (Figure 1a), are close to ~100 μm and ~300 μm, respectively. In the case of Cu/SS sheets, a non-regular interface in this section is found to exist. On the contrary, the Cu/SS interface in the $ND/DD^1$ section shows a regular waviness with the wave amplitude and the wave period close to 450 μm and 950 μm, respectively (Figure 1c).

The wave formation coincides with the formation of solidified melt zones that are preferentially located at the wave crest and within the wave vortexes. The cracks within the solidified melt zones, commonly observed in other metal combinations, e.g., [4,7,9] were only occasionally detected in solidified melt zones formed at both interfaces of the Ta/Cu/SS clad. However, if observed, they were always limited to the zone of solidified melt and they propagated perpendicularly to the interface between solidified melt and pure metal. None of these cracks have shown any tendency to propagate across the base materials.

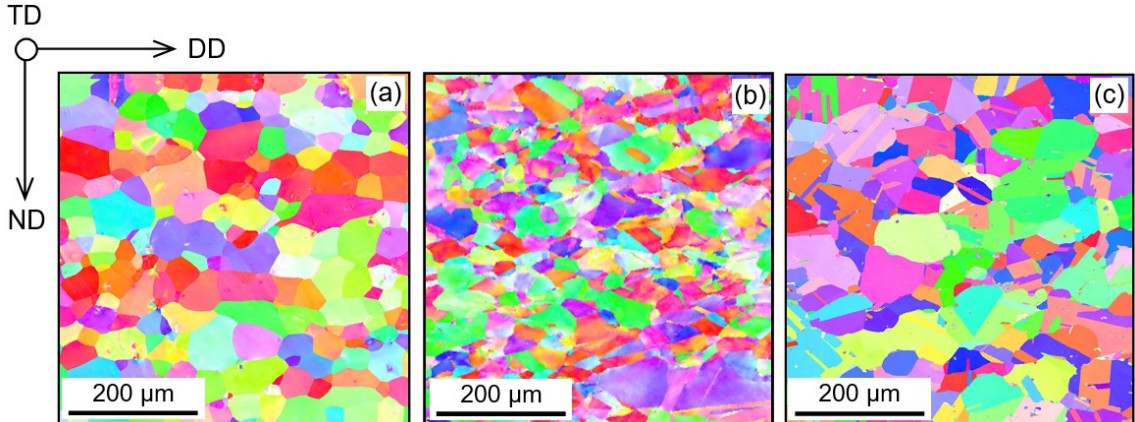

**Figure 2.** The initial microstructure of (**a**) tantalum, (**b**) copper and (**c**) stainless steel sheets taken in the normal/rolling direction (ND/RD) plane. Scanning electron microscopy/electron backscatter diffraction (SEM/EBSD) images of local orientation measurements with a step size of 200 nm.

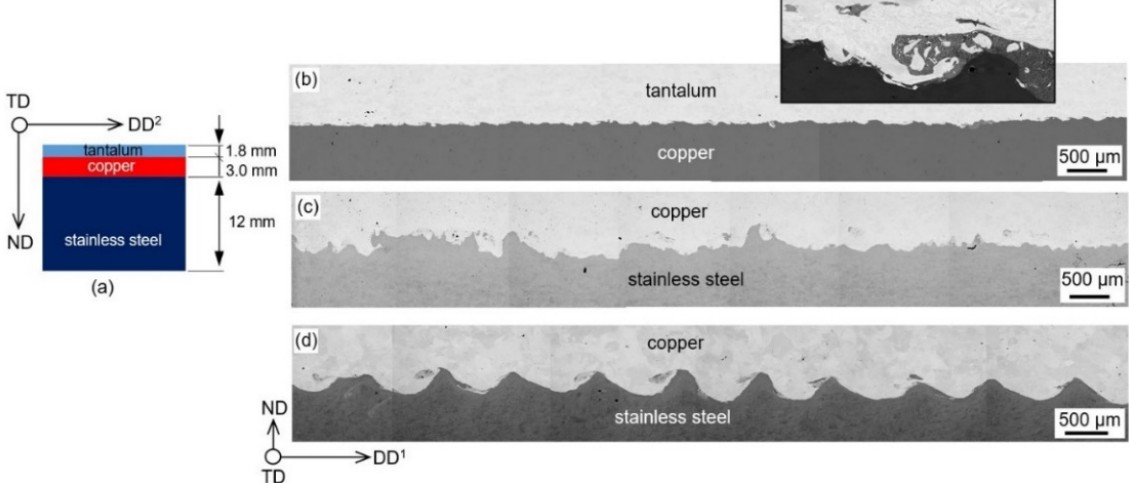

**Figure 3.** (**a**) Thickness of initial sheets, and the wavy interfaces, (**b**) regular wavy interface between Ta and Cu and (**c**) non-regular interface between Cu and SS observed in a final clad along ND/DD$^2$ section. (**d**) The regular wavy interface between Cu and SS observed along the ND/DD$^1$ section.

### 3.2. Microstructure of Severely Deformed Layers of Parent Sheets Near the Interface—SEM/EBSD Analysis

Figures 4 and 5 present the inverse pole figures (IPF) orientation maps combined with the image quality (IQ) factor. The IQ component emphasizes grain and interphase boundaries. The analysis is supported by direct SEM/EBSD/EDS chemical composition determination. The SEM/EBSD maps of the vortex region show a severely deformed microstructure of parent sheets composed of elongated cells/(sub)grains with a tangled network of dislocations (Figure 4a). The points inside the zone of solidified melt are mostly indexed as 'pure' metals with some quantity of not or mis-indexed pixels. Further chemical analysis reveals that the solidified melt zones contain both elements in the vortex region.

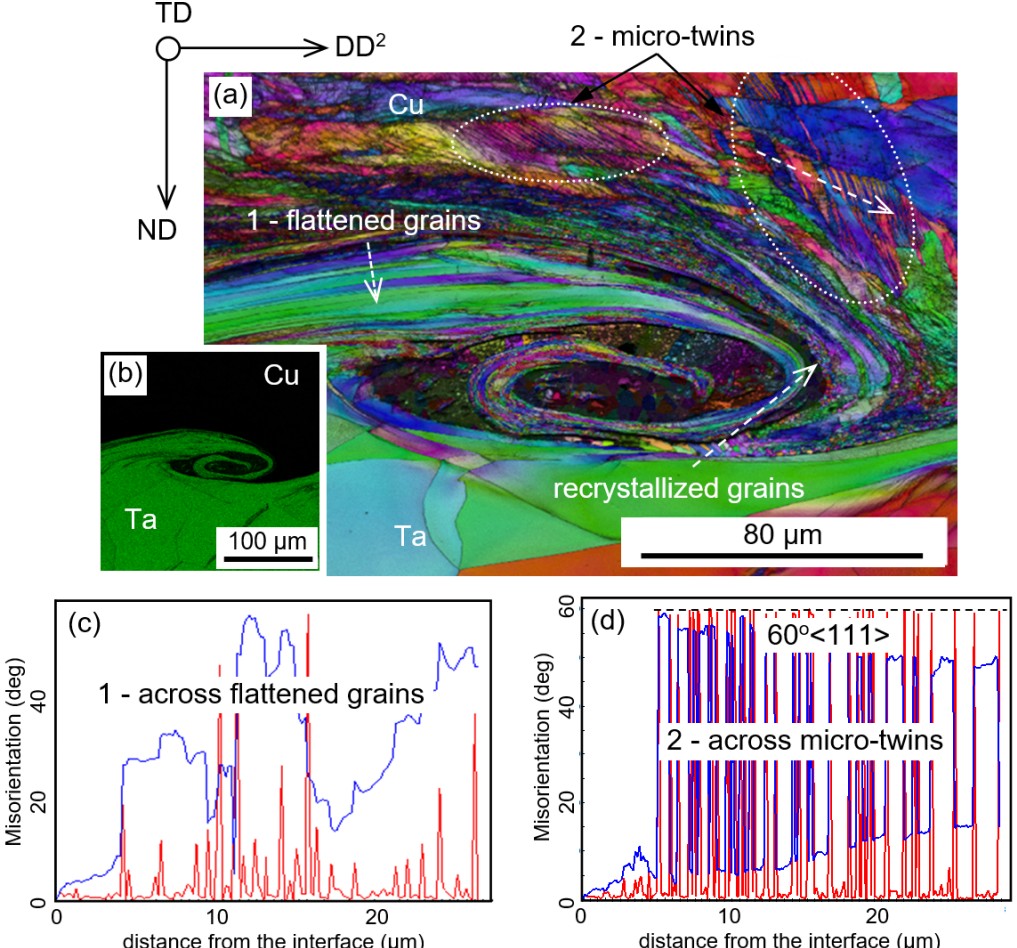

**Figure 4.** (**a**) Orientation map showing a typical vortex region close to the Cu/Ta interface, and corresponding (**b**) chemical composition map showing the distribution of Ta. Misorientation line scans across (**c**) flattened grains in Ta, and (**d**) clusters of twins in Cu. SEM/EBSD measurements with a step size of 100 nm along the ND/DD$^2$ section.

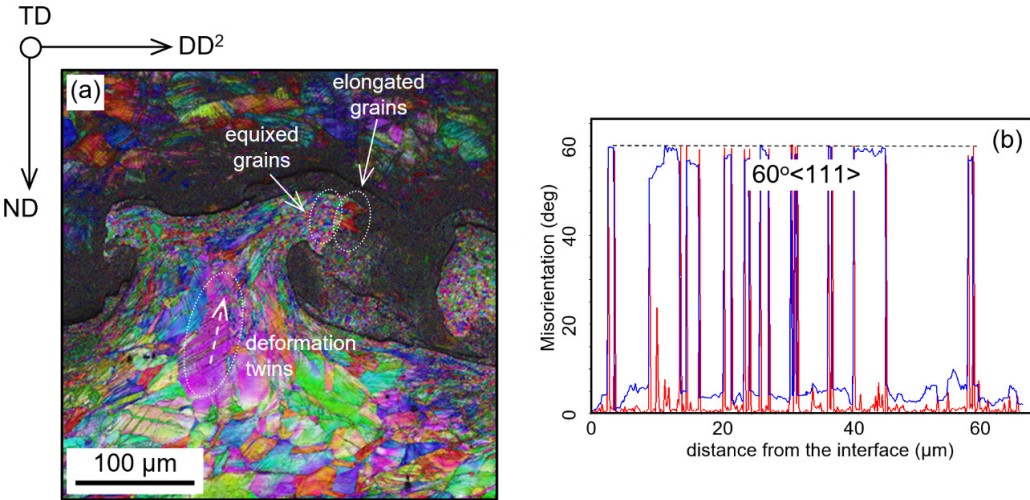

**Figure 5.** (**a**) Orientation map showing typical vortex near the SS/Cu interface along ND/DD$^2$ section. (**b**) Misorientation line scan across the cluster of twins. SEM/EBSD measurements with step size of 100 nm.

### 3.2.1. Ta/Cu Interface

The severely deformed layers of tantalum are relatively narrow and limited to the thickness of a few tens of microns. In contrast, in the copper sheet a significantly larger strain hardened layer is observed (a few hundreds of microns). For instance, Figure 4a shows the microstructure of the interfacial layer around and inside a wave vortex. The formation of vortexes leads to a characteristic interlock microstructure between tantalum and copper, as is clearly presented in Ta chemical composition map (Figure 4b). Interfacial layers of both parent sheets are composed of flattened grains with a thickness of 1–2 μm. Grain to grain misorientation plot shows that to a large extent the flattened grains are separated by high angle boundaries (Figure 4c). The orientations of neighboring, flattened grains are nearly symmetrical. Such a grain shape and misorientation relationship between grains refer to plane strain conditions and deformation banding mechanism dominating in medium-to-high stacking fault energy fcc or bcc metals [38]. However, in some places these flattened grains undergo intense recrystallization; this leads to the formation of layers composed of very small (<1 μm) equiaxed grains. Another characteristic feature of the as-deformed structure of copper (but not present in Ta sheet) is massive deformation twinning. This mechanism is supposed to accommodate the strains that emerged in the vortex formation. Figure 4d shows the misorientation angle distribution along the pathway marked by an arrow in Figure 4a. The misorientation angle between neighboring platelets is 60° with the misorientation axis corresponding to the <111> direction. This clearly indicates that the micro-twins formation is similar to the periodic twins that have been observed in pure copper deformed at extremely high strain rates, as observed earlier by Crossland and Williams [39] and by Lee et al. [17] in EXW copper to copper sheets. The line scan across the deformation twins can be contrasted with the misorientation vs. distance line scan presented in Figure 4c, where the opposite tendency of crystal lattice rotation in the neighboring layers (flattened grains) is observed. However, the misorientation angles between the layers always display values lower than 60°.

In the present work the most regions inside the solidified melt zones near the Ta/Cu interface, were indexed as 'pure' Ta or Cu, with some quantity of not or mis-indexed pixels [11,30,31].

### 3.2.2. SS/Cu Interface

Figure 5a shows an SEM/EBSD image with a vortex region along the SS/Cu interface. In ND/DD$^2$ section, unlike the typical wavy shape, the SS/Cu interface resembles a column capital. There are a molten and re-solidified regions near the vortices, marked by very low Kikuchi contrast (dark grey or black pixels). Poor indexing of those areas can be directly related to the formation of non-equilibrium ultra-fine grained (or even amorphous) phases based on Cu and elements present in stainless steel. Orientation maps show that interfacial layers of both parent sheets are severely deformed with a significantly refined microstructure. The interfacial layers of the SS sheet are composed of grains with the size ranged between 0.5 μm and 2 μm. The width of this layer is 50 μm–100 μm. However, at larger distances from the interface, much coarser and only slightly deformed grains can be observed. In the Cu sheet, the as-deformed grains are not exclusively observed near this interface. In areas situated near the wave crests and adhering to solidified melt zones, a more uniform structure of equiaxed grains (free of dislocations) with an average grain size of about 1–3 μm is observed. This strongly indicates intense recrystallization. In some places, this structure evolves into columnar grains indicating the occurrence of oriented grain growth along the preferred heat flow direction. The large columnar grains grow perpendicular to the melted zone/copper interface. The length of the larger axis of columnar grains is between 5 μm and 15 μm. However, in some of the Cu grains situated inside the vortex (pillars) region, the occurrence of dislocation slip is accompanied by intense deformation twinning. This is confirmed by a misorientation line scan in Figure 5b indicating compact clusters of deformation twins, similar to those observed in Cu near the Ta/Cu interface. The equilibrium phases corresponding with the Cu and Fe(Cr, Ni) binary diagrams were identified only accidentally near the Cu/SS interface. This observation confirms the early hypothesis [4,16] that the melted zones are mostly composed of a mixture of pure parent metals.

### 3.3. Dislocation Structure of Parent Sheets—TEM Analysis

The dislocation structures of interfacial layers of parent sheets were analyzed in areas of the wave valley, where large zones of solidified melt are not observed at the optical microscope or low magnification SEM scale. Generally, the structures were composed of fine equiaxed or elongated cells/(sub)grains with an increased density of dislocations inside them. Such microstructures are typical for strain hardened materials with a tangled network of dislocations, high vacancy concentration and possibly large numbers of microtwins (in the Cu and SS).

The most characteristic feature observed in the Ta sheet in layers near the Ta/Cu interface is the structure composed of relatively wide microbands (Figure 6a), whereas in the Cu sheet (at the same interface) the formation of extremely fine but equiaxed (sub)grains (Figure 6b) is observed. In layers near the Cu/SS interface the structure of Cu sheet is significantly less deformed as compared to the layers of Cu sheet situated near the Cu/Ta interface. Despite a huge shear strain, the structure of initial grains with recrystallization twins is still apparent (Figure 7a). The microstructure of SS plate in layers adjacent to the Cu sheet is composed of fine (sub)grains with a diameter of a few hundreds of nanometers (mostly <500 nm). A large number of dislocations accommodated in the cells/(sub)grains of both metals is a strong indication that the deformation processes are prevailing over the thermally activated softening ones, i.e., recovery and recrystallization, as suggested earlier for Al/Cu [9], Zr/(carbon steel) [10] and Al/Ti [14] clads.

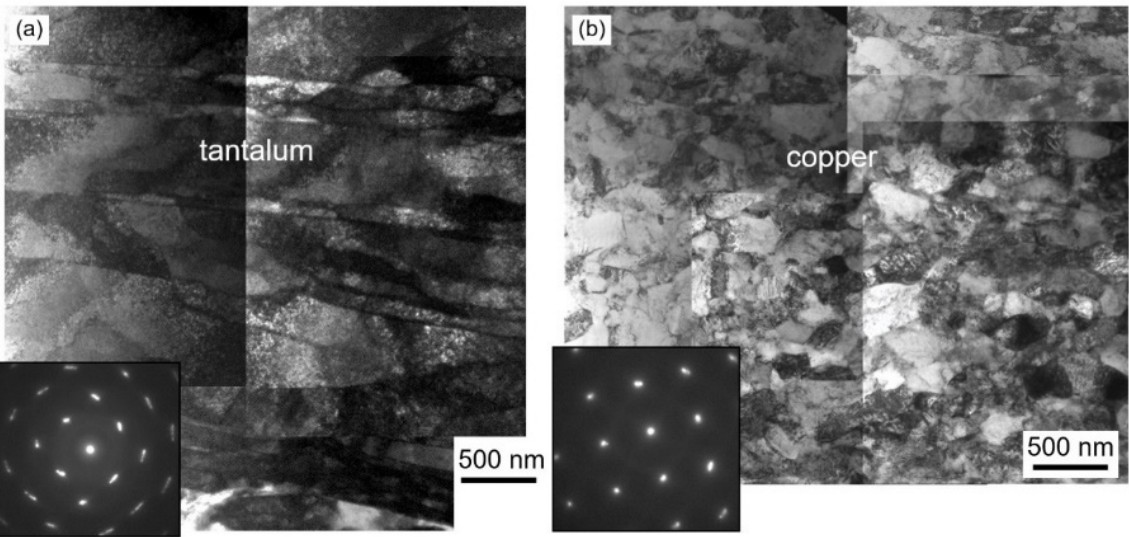

**Figure 6.** Transmission (TEM) bright-field images showing the structure composed of (**a**) elongated cells in Ta and (**b**) fine grains in Cu.

The wave valleys do not contain any macroscopically visible solidified melt zones (Figure 7a–c). However, a very thin layer of solidified melt, of few tens of nanometers is identified at large magnifications, as presented for Cu/SS interface in Figure 7d. This confirms the early thesis that the presence of a very thin reaction layer is one of the most important factors that guarantee a good bond between welded sheets and improves delamination resistance, as suggested in [4,8,10].

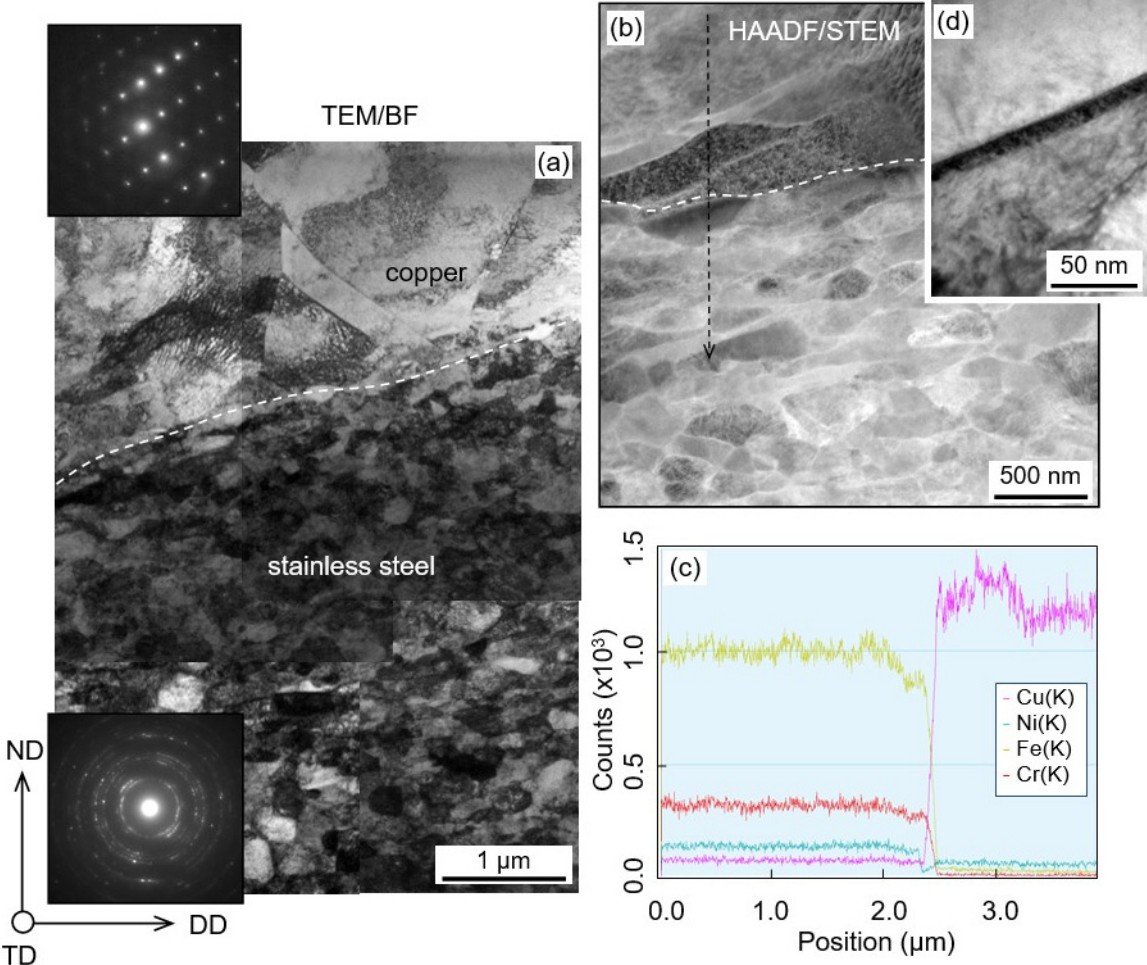

**Figure 7.** (**a**) TEM bright field and (**b**) TEM/High Angle Annular Dark Field (HAADF) images showing microstructure of copper and stainless steel near the Cu/SS interface. (**c**) Chemical composition line scan across the Cu/SS and (**d**) interfacial region observed at high magnification documenting the presence of a very thin layer of solidified melt.

## 3.4. Phase Constitution inside Solidified Melt Regions

The most spectacular microstructural changes are observed in the solidified melt zones. The melt zones may exhibit a different form—starting from large, nearly equiaxed ones, going through thin, and ending up with extremely thin layers. They are situated inside the wave vortexes (as inclusions inside the Ta or Cu sheet) and at the crest or on the bottom part of the wave (extremely thin layers). Large solidified melt zones situated inside the wave vortexes can be (i) entirely surrounded by Ta or Cu, without any contact with Cu or SS, respectively, or they can be (ii) still partly adjacent to the neighboring sheet (Cu or SS).

Some examples of solidified melt zones of the second type are presented in Figure 8a,b. These SEM images taken with a backscattered electron detector (SEM/BSE) show sharp interfaces between the zone of solidified melt and parent materials. This implies significant changes in chemical composition across the boundaries. Additionally, the areas of pure metals (mostly of a higher melting point) are detected inside the solidified melt.

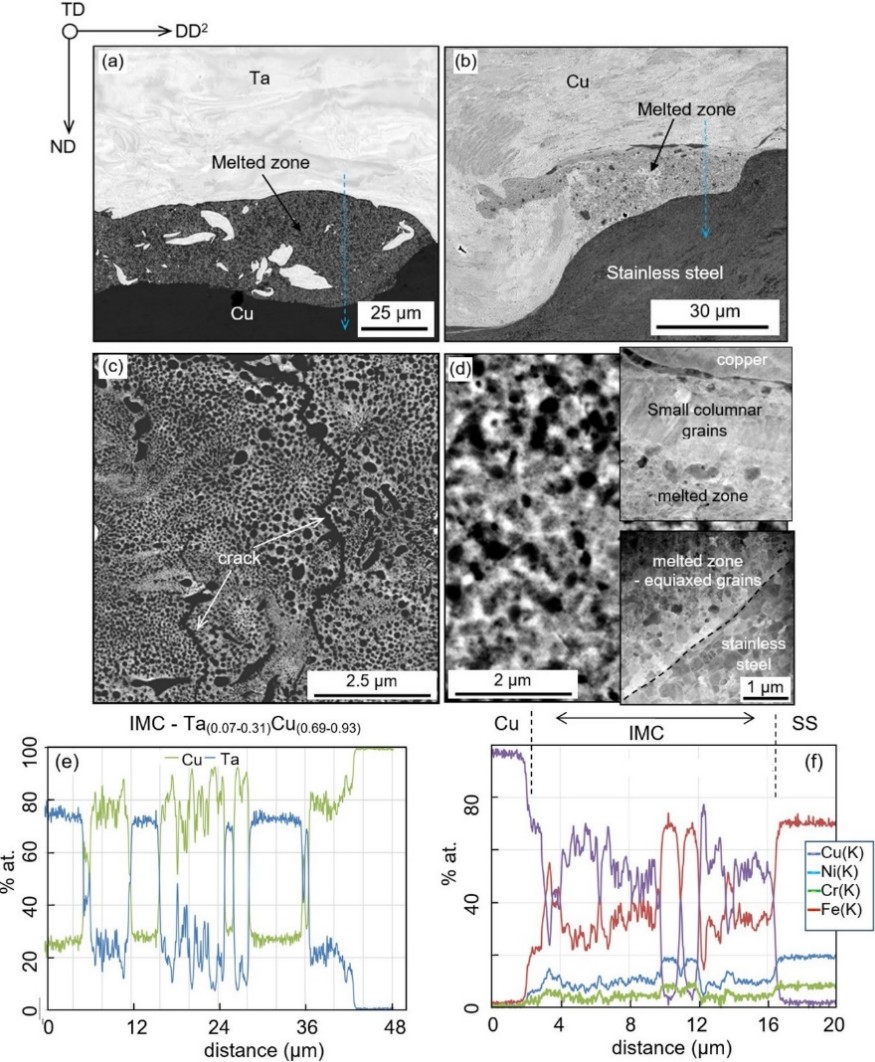

**Figure 8.** Microstructure of solidified melt zones formed between (**a**) Ta and Cu, and (**b**) Cu and SS, (**c**,**d**) details of internal microstructure. (**e**,**f**) Chemical composition line scan showing the distribution of main elements in the solidified melt region.

### 3.4.1. Chemical Composition of Solidified Melt Zones Formed between Tantalum and Copper

The internal microstructure of solidified melt zone shows small areas of alternating BSE contrast. For instance, Figure 8a depicts an enlarged part of the vortex between Ta and Cu sheets. In some cases, the pure metals reveal rotational character due to the vigorous stirring and mixing of the material in the liquid and/or semi-liquid states. A more detailed analysis shows that the internal structure of the solidified melt region can be defined as a mixture of fine Cu and Ta particles of different sizes. Figure 8c shows an SEM/BSE micrograph with a typical structure of the solidified melt. In most observed cases the size of spherical particles of Ta is ranged between 10 nm and 500 nm and they are homogeneously distributed in the Cu matrix. Nevertheless, there were experimentally discovered micro volumes in which the matrix material is Ta and Cu is in the form of compact spherical aggregates. However, as shown by Bataev et al. [14] and Parchuri et al. [32] they may coexist with $Ta_xCu_{1-x}$ based intermetallics and decagonal quasicrystals at the Ta/Cu interface.

Despite the fact that the chemical composition inside the zone of solidified melt varies practically from 0% to 100% of a given element, a strong preference of chemical composition close to $Cu_{0.25}Ta_{0.75}$ and $Cu_{0.75}Ta_{0.25}$ is observed. This can be clearly seen in Figure 8e in the chemical composition line scan along the blue dashed arrow marked in Figure 8a.

### 3.4.2. Chemical Composition of Solidified Melt Zones Formed near the Stainless Steel and Copper Interface

In comparison to Ta/Cu interface, the grains inside the solidified melt zone near the Cu/SS interface show different morphology and chemical composition (Figure 8b,d,f). Most of the grains crystallize in the form of small dendrites. However, the grains that nucleate just near the Cu sheet form a characteristic sublayer composed of small columnar grains with longer axis perpendicular to boundary between solidified melt and copper. This indicates a more effective heat transfer across the interface towards Cu as compared to the other side of the melted zone, where only ultra-fine, but nearly equiaxed grains are formed (see insets in Figure 8d). Nevertheless, large volumes of the melted zone are occupied by ultra-fine-grained phases. Various chemical compositions are identified inside the zone. To a larger extent the phases are enriched in copper (ranging between 60% and 90%) (Figure 8f).

The solidified melt zone can also be enclosed inside the parent metal. Such an example is shown in Figure 9 where a solidified melt is surrounded by the Cu matrix. The corresponding chemical composition maps show the distribution of Cu and the main elements of stainless steel (Fe, Cr and Ni). The content of elements is quite similar to that observed inside the open zones and is close to $Cu_{(0.60–0.90)}Fe_{(0.20–0.72)}Cr_{(0.05–0.19)}Ni_{(0.02–0.08)}$. The maps also revealed large fragments of steel enclosed inside the solidified melt zone.

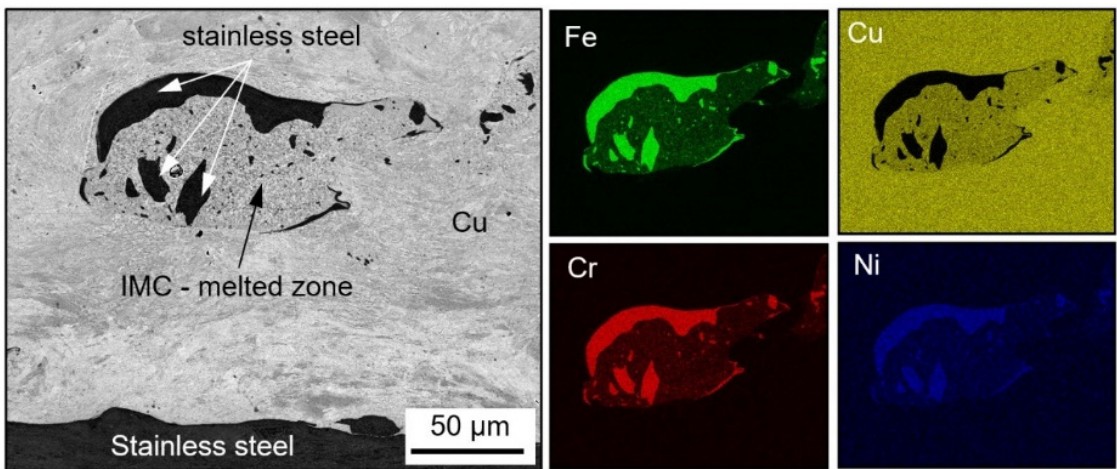

**Figure 9.** SEM/backscattered electron detector (BSE) image showing solidified melt zone enclosed inside the deformed Cu and corresponding chemical composition maps documenting the distribution of Cu and main elements in stainless steel.

### 3.5. Microhardness across the Interface

In order to correlate the mechanical properties with the corresponding changes in microstructure the microhardness measurements were performed. The microhardness profiles along ND (in the ND/DD$^2$ section) in areas near the wave crests are presented in Figure 10a,b. They show the distribution of the strain hardening across the interface. The average microhardness values of the base materials in the fully recrystallized states, i.e., before cladding, are marked as blue horizontal dashed lines. For both interfaces, the microhardness of parent sheets strongly increases in layers near the interface, whereas at the center of the sheets does not differ significantly from those of the initial state (Figure 10a,b). It is apparent that the most radical increase of strain hardening is observed in the interfacial layers of SS sheet near the SS/Cu interfaces (Figure 10a,b). However, the microhardness values inside the solidified melt zones strongly scatter. For both interfaces, the microhardness of solidified melt zones reached values between 240 and 370 HV (Figure 10c) and between 210 and 260 HV (Figure 10d) for SS/Cu and Cu/Ta interfaces, respectively. These values significantly differ from those obtained for

SS/Ta metals composition, where the microhardness of solidified melt regions ranged between 720 HV and 1080 HV [14]. This is twice or even three times more than those observed in strongly refined (and strain-hardened) layers of SS sheets. However, in that case, a strong scattering of values was also observed.

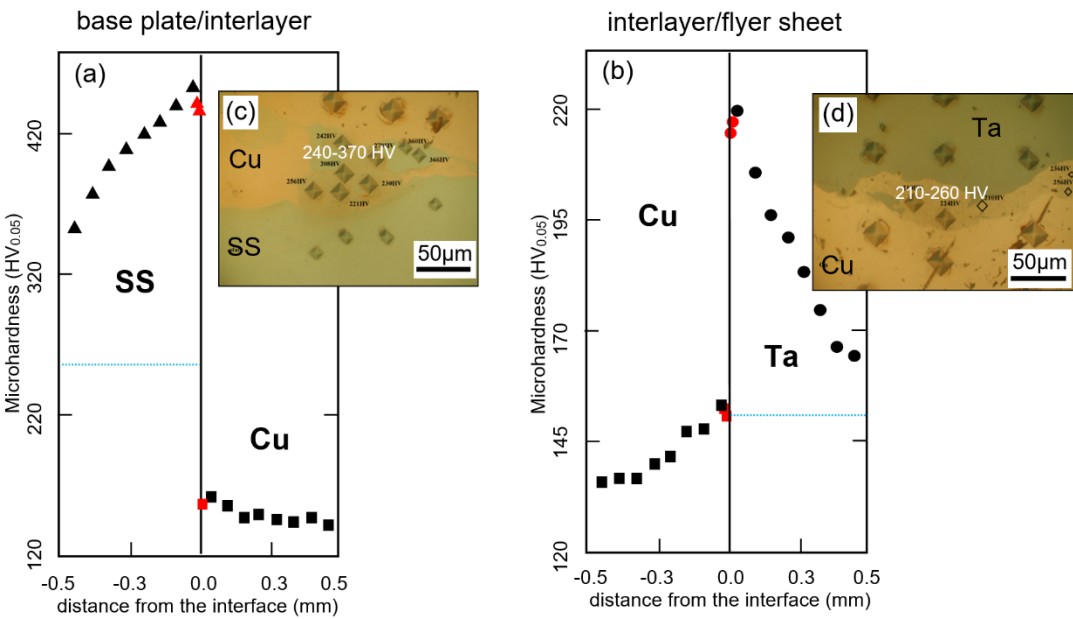

**Figure 10.** Vickers microhardness across the interfacial layers in (**a**) SS/Cu and (**b**) Cu/Ta clads, (**c**,**d**) corresponding optical micrographs showing values of microhardness of solidified melt zones. Red triangles (**a**,**b**) indicate the values of microhardness measured in the interfacial layers close to a large melted zone. Microhardness of copper in a fully recrystallized state is 110 HV.

As a next step, the lateral bending test was performed to evaluate the influence of the interfacial microstructures on the integrity of the joints. The tested specimens were deformed up to 90° bending angle (Figure 11). No tearing, fracture, or separation of the sheets was observed. The cracks, which were originally localized within the solidified melt region start to expand into the stainless steel substrate upon loading, whereas there was no propagation of the pre-existing cracks into the copper, stainless steel or tantalum sheets. This shows that Ta/Cu/SS welding is fully applicable in service even in a bent form.

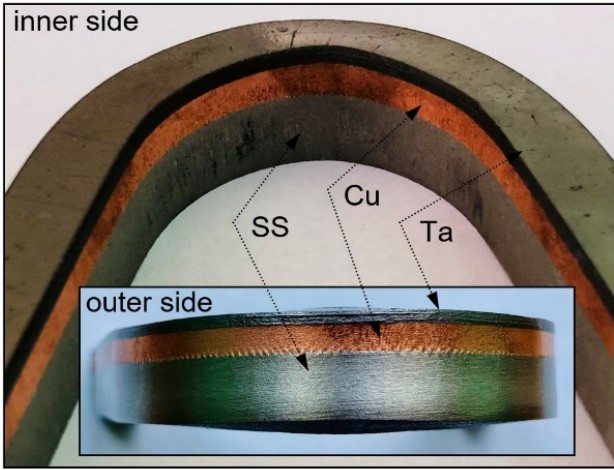

**Figure 11.** Optical micrographs showing macro-structure of the sample after (lateral) bending test. No delamination of the clad components and macro-cracks propagation is observed.

## 4. Discussion

This study describes several important aspects of our understanding of EXW and the evolution of microstructure in the process of joining. These changes are twofold. The first ones are due to strain hardening, recovery, and recrystallization of interfacial layers of the parent sheets. The latter ones are due to the phase transformations in the solidified melt zones due to fast heating followed by extremely fast cooling.

### 4.1. Microstructural Changes in the Parent Sheets

The strain hardening processes in the parent sheets prevail over the thermally activated softening ones in areas far from the large melted zones. As a consequence, the presence of strain-hardened structures can be regarded as a factor that increases the strength of the clad. However, in layers adhering to large melted zones the softening due to recrystallization is noticeable, as observed earlier for other metal compositions, e.g., [4,7,14,40]. The recrystallization starts to occur just during clad formation as a result of heat transfer between the large melted zones and severely deformed layers of parent sheets/plates. In areas near the interface, this process leads to the transformation of (sub) grains with randomly distributed dislocations into equiaxed fully recrystallized grains free of dislocations. Since the highest temperature is reached at the interface, it is often observed that these areas undergo intense recrystallization or even abnormal grain growth. This, in many cases leads to the situation where the average size of grains situated at the interface is larger than that of more distant from the interface, as also presented by Chu et al. [18] for Ti to carbon steel clad. As a consequence of the recovery and recrystallization processes, a pronounced drop of microhardness in layers of parent sheets directly adhering to the large melted zone can be detected.

### 4.2. Microstructural Changes due to Formation of the Solidified Melt Regions

The history of dramatic temperature changes during the impact process can be used to elucidate the microstructure evolution of the solidified melt region. During the cooling period (as the collision point is moved forward), the cooling rates calculated by various authors always exceed the $10^5$–$10^7$ ks$^{-1}$ range, e.g., [10–12,18]. These rates make the thermodynamic conditions very similar to that observed during spin melting, where the formation of extremely fine-grained or even amorphous phases is expected.

Although a low microhardness has been observed in the solidified melt zone (as compared to strain hardened layers of 304 L SS sheet), upon bending the strain is mostly accumulated by the fully recrystallized microstructure of Cu that surround the solidified melt region. It is clear that this behavior is quite different from those observed for other metal compositions, e.g., based on steels and reactive metals, where the formation of brittle solidified melt zones of high hardness is dominant, e.g., [4,7,10].

For the particular case of Ta/Cu metal composition, a possible mechanism of intermediate heterophase region formation was proposed by Maliutina et al. [31]. In this mechanism there is not necessary to exceed the melting point of Ta since the globular particles of Ta are plunged into solidified Cu. However, it is very difficult to state without any doubt, whether this mechanism is the only one or just dominant. The weak point of the mechanism proposed in [31] is that the matrix is not pure copper but is mostly a solid solution of Ta in the Cu (see Figure 8). However, aside from the limited information on the phase constitution in the Cu-Ta immiscible systems, the formation of metastable phases can be expected due to rapid cooling during solidification of melted volumes, as described for other processes in [33–37,41,42]. Due to the extremely high dynamic of the EXW process, there is no time for the diffusion in the solid state, hence the melting of Ta can be expected. However, to solve this problem further analyses with the use of TEM are needed.

### 4.3. Microstructure vs. Strength Properties

An important question then arises here is—how the processes occurring inside the deformed layers and melted regions can influence the strength of the clad? Although, the strain hardened layers of parent sheets exhibit high hardness, the recrystallized structure near the interface and cracks propagation inside the solidified melt zone can countervail the overall strengthening effect.

The presence of ultra-fine-grained phases inside the solidified melt zones improves the clad strength, but on the other hand, it may lead to an increase in the resistance of a material to plastic deformation. The cracks appear inside the solidified melt zones owing to the shrinkage during the solidification of molten materials which significantly decreases the clad strength. However, in Cu–Ta and Cu–SS solidified melt regions the cracks are observed only accidentally. During the bending test some newly generated small interfacial discontinuities appear and grow together with pre-existing ones only inside the Cu–Fe(Ni, Cr) solidified melt regions. However, the crack propagation towards parent materials was not observed. The strain hardening of the interfacial layers of parent sheets can be attributed to the increase of lattice defects, such as dislocations, vacancies, (sub)grain boundaries, and in more global scale intense deformation twinning. However, the recovery and recrystallization processes, initiated just during EXW can decrease the density of structural defects due to heat transfer from large melted zones towards severely deformed layers [43]. This, in turn, leads to a decrease of strength.

This study also contributes to the issue that cannot be completely solved by optical microscopy and low magnification of SEM analyses. We are referring here to the mechanisms responsible for bond formation. Generally, there are two interpretations strongly disputed in the literature. On the one hand, high-resolution TEM studies of similar [8] and dissimilar [4,10,12] metals show the formation of an extremely thin layer of solidified melt of a few tens of nanometers in thickness, between neighboring sheets. The formation of a very thin layer of solidified melt, as observed in Figure 8, was predicted by Carpenter [44] who suggested that the weld interface is in reality a thin melted layer. The open question is—if this is a requirement for all bonding conditions, or if it is a result of excessive explosive loading? However, in numerical modeling of EXW of Cu to Cu, Lee et al. [17] argue that an essential mechanism of bonding is the jetting and extremely high pressure which are responsible for squeezing two sides free of impurities to atomic contact. Despite this mechanism was used many times in the past by various authors, this study tends more to metallurgical bonding. This melting and mixing can give a continuous reaction layer between the matching pair.

This study also points to open questions and opportunities. The presence of a very thin (<300 nm) reaction layer along with the entire interface, free of cracks, can play a decisive role in enhanced resistance to delamination and may be responsible for proper bonding between the sheets. Although the appearance of a thin reaction layer is typical for most of the clads, it is difficult to state without any doubt if the presence of a thin intermetallic layer is the necessary condition for proper bonding or if it only supports (as suggested by some of the authors) the essential mechanism based on the jetting and squeezing two sides to atomic contact.

## 5. Summary

The present work describes the microstructure of the Ta/Cu and Cu/SS interfacial layers of explosively welded Ta/Cu/SS clad. It is shown that the hardening and softening processes are strongly related to the microstructure and clad strength evolution. The processes that increase the yield strength of the material are severe plastic deformation, the formation of thin layers of ultra-fine grains, and solidified melt zones, while dynamic recovery, recrystallization, and crack formation inside the solidified melt cause softening.

SEM/EBSD analyses revealed a complex microstructure of parent sheets, which consists of characteristic features such as interlock microstructures, elongated grains, or twins. Strain hardening processes predominate softening ones in the interfacial regions far from large solidified melt zone. It has been established that a high-rate shearing of interfacial layers leads to a strain-hardening due to

intense slip and deformation twinning in copper sheets at both interfaces. In layers surrounding the melted zones, the as-deformed grains are replaced by new recrystallized (equiaxed or columnar) ones due to the heat transfer from the melt to the severely deformed metals. The microhardness of welded sheets increases significantly as the joining interface is approaching excluding the volumes directly adhering to large melted zones, where a noticeable drop of microhardness due to recrystallization is observed.

However, precise analysis of dynamic recrystallization and recovery is exceptionally complicated as both phenomena strongly depend on both temperature and strain distributions near the interface as well as the melting point of the joined metals. Another observation is a quite different morphology of the solidified melt zones at the Ta/Cu and Cu/SS interfaces. The melted zones close to the Ta/Cu interface consist mainly of a mixture of pure Cu and Ta particles of different sizes, whereas the melted zones near the Cu/SS interface are composed of nano-grained compounds based on elements of both neighboring joined sheets. The microhardness of solidified melt regions formed near both interfaces is significantly lower than that measured inside the strain hardened layers of stainless steel. Therefore, the integrity of all clad components is fully conserved on lateral bending.

**Author Contributions:** Conceptualization, H.P.; methodology, H.P. and R.C.; formal analysis, H.P.; investigation, R.C. and I.M.; writing—original draft preparation, H.P.; writing—review and editing, R.C.; visualization, H.P.; project administration, I.M; funding acquisition, H.P. All authors have read and agreed to the published version of the manuscript.

**Funding:** This research was funded by the National Centre for Research and Development of Poland under grant TECHMATSTRATEG2/412341/8/NCBR/2019 (EMuLiReMat).

**Conflicts of Interest:** The authors declare no conflict of interest.

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
