# Peer review of "Structural Properties of Interfacial Layers in Tantalum to Stainless Steel Clad with Copper Interlayer Produced by Explosive Welding"

_metals, doi:10.3390/met10070969_

Round 1
Reviewer 1 Report
Dear Authors,
I read your manuscript with great attention, because explosive welding is close to my scientific interests. Solid state welding processes, including explosion welding, are one of the main research trends in modern welding technology.
The submission is an experimental work, which concerns the analysis of the structure and properties of two intermediate layers of an explosion welded joint: Ta/Cu/SS. I have no doubt that the topic presented in the article is important and current. The presented results of advanced metallographic research are described in detail and commented on taking into account information from the literature. The given results will be an important source of information for industrial engineers and scientists, because they give a lot of basic data.
On the technical side, I rate the preparation of the manuscript as high, the work is written professionally.
Below are the remarks and comments that came to my mind when analyzing the content of your submission.
The biggest disadvantage of the manuscript is the lack of information about the materials: What grades of stainless steel/copper/tantalum were used? This information should be given in the abstract, keywords, Introduction and Chapter 2. I suggest adding trade names and table with chemical composition.
In the abstract and in the manuscript text the authors use the term 'parent'. However, current welding terminology guidelines recommend the word "base".
In the Introduction chapter I advise you to add the names of specific applications of such joints.
Chapter 2: The description of the explosion welding technology is very accurate. Is it possible to make such a joint in one stage of welding? If not, I suggest you add that this is impossible because, for example, the explosion welding windows for both joints are separable and exclude it.
Line 112: correct: “micro hardness”.
Change “Fig.” to “Figure” throughout the text.
Format References according to journal guidelines.
Line 436: reference [36] should be [37].
Author Response
Changes made to the paper " Structural properties of interfacial layers in tantalum to stainless steel clad with copper interlayer produced by explosive welding ".
We should like to thank the Reviewer for providing a wider perspective to the work and making useful suggestions to improve the paper. The detailed response to Reviewers comments is as follows: Reviewer 1
• The biggest disadvantage of the manuscript is the lack of information about the materials: What grades of stainless steel/copper/tantalum were used? This information should be given in the abstract, keywords, Introduction and Chapter 2. I suggest adding trade names and table with chemical composition.
It has been corrected. Table 1 showing chemical compositions of joined materials is now introduced. • In the abstract and in the manuscript text the authors use the term 'parent'. However, current welding terminology guidelines recommend the word "base".
Indeed, for the description of the clad formation process the terms ‘flyer’ and ‘base’ sheet/plate are always used. However, the term ‘parent sheet’ is used to underline the events that occur in the explosively welded materials of both flyer or base sheets/plates but not inside the solidified melt regions which are the mixture of elements from both neighbouring sheets/plates. This term is especially used for the description of processes such as severe plastic deformation, mechanical twinning and/or recovery and recrystallization that take place in both sheets. To make more clear an additional explanation is given. • In the Introduction chapter I advise you to add the names of specific applications of such joints.
Few sentences indicting application fields of Ta/SS metals compositions has been added in the Introduction section, see below.
It is well-known that Tantalum is the ideal metal for most severe corrosion situations, especially at high temperatures. It is capable to protect all installations exposed to highly oxidizing or caustic environments, where glass lined equipment is subject to mechanical damage or thermal shock failures. Since the high cost of tantalum has traditionally been a major impediment in wide-scale industrial applications, such as large pressure vessels, therefore, it is advisable to use them rather as a coating on carbon, stainless or ‘duplex’- type steels. According to corrosion studies, even a 100 μm tantalum coating provides excellent corrosion resistance for the base metal. Compound materials of this type are both structurally efficient and economical; the tantalum cladding surface offers all of the benefits of wrought tantalum, whereas the steel base metal is low cost and readily fabricable.
Since, further processing of bi-layered Ta/SS composites is strongly restricted (due to limited heat transfer from the heat-affected zone), therefore, to solve the problems associated with conventional welding of the Ta/SS clads an intermediate layer, made of soft material and high thermal conductivity, e.g. copper, to joint two sheets is used. - Chapter 2: The description of the explosion welding technology is very accurate. Is it possible to make such a joint in one stage of welding? If not, I suggest you add that this is impossible because, for example, the explosion welding windows for both joints are separable and exclude it.
It is possible to fabricate the Ta/Cu/SS composite in one act. However, in the analysed case the clad was manufactured in two steps, since we used typical clad for electrotechnical applications (as a base plate) covered by Ta. We produced relatively thick interlayer in order to analyse deformation processes inside the copper sheet. Especially, the effect of high strain rates on the deformation mechanism via twinning and shear banding. Line 112: correct: “micro hardness”.
Change “Fig.” to “Figure” throughout the text.
Format References according to journal guidelines.
Line 436: reference [36] should be [37].
It has been corrected.

Reviewer 2 Report
Authors presented an extensive study of the mechanical properties tri-metallic sheet obtained through explosive welding. The paper is characterized by a good quality and I have no particular remarks.
However, the paper is especially focused on the microstructural properties of the joint, while in the introduction section authors spoke about "bending strength". In reality they carried out a bending test and observed qualitatively the absence of cracks. Therefore, I suggest to correct the paper avoiding to use the word "strength", which effectively has not been determined.
Author Response
Reviewer 2
Authors presented an extensive study of the mechanical properties tri-metallic sheet obtained through explosive welding. The paper is characterized by a good quality and I have no particular remarks. However, the paper is especially focused on the microstructural properties of the joint, while in the introduction section authors spoke about "bending strength". In reality they carried out a bending test and observed qualitatively the absence of cracks. Therefore, I suggest to correct the paper avoiding to use the word "strength", which effectively has not been determined.
It has been corrected accordingly.

Reviewer 3 Report
The manuscript presents the experimental results of analysis of microstructure of interfacial layers in Ta/Cu/SS composite produced by explosive welding. The authors apply different modern methods (SEM/MEM, EBSD) of microstructure analysis. The technique of explosive welding of Cu/SS and Cu/Ta composites known and presented in the scientific papers. Any commercial clad products are available in the market. The paper is consistent, but information is interesting mostly for industrial practice and for confirmation of known facts. The combination of Ta/Cu/SS is not the new idea. T. Balzynski talked about such clad plate in the monograph (https://doi.org/10.1007/978-94-011-9751-9). Bouckaert with co-authors analyzed the similar products in monograph (Bouckaert, G.P; Hix, H.B.; Chelius, J. (1974) Explosive-bonded tantalum-steel vessels. DECHEMA Monograph). But authors not show, that microstructure of interfacial and vortex regions of Cu/Ta, Cu/SS and Ta/Cu/SS composites already studied before and presented in the articles. For example: doi:10.1016/j.matdes.2004.07.021; doi: 10.1016/j.matdes.2018.05.027; doi: 10.1080/09276440.2020.1736880; doi: 10.1080/10426914.2012.736665; doi: 10.1109/IFOST.2014.6991157; doi;10.1016/j.matdes.2019.108348
Therefore, the reasons of this research and review of published articles must be improved and related with object of this work and published research results. Authors must explain more detail the main reasons of this study and novelty of presented experimental results, compare them with published information, extend the analysis and comments of main results.
The chapters 4.2 and 4.3 present the information, which is actual for introduction or literature overview chapter. Authors explain but not discuss about achieved microstructural analysis results. The figure 1 is picture of typical for explosive welding technology. There is no big reasons to put such picture in the article. Authors must to revise and improve the summary and general conclusions taking into account main findings.
Author Response
Reviewer 3
The manuscript presents the analysis of experimental results of interfacial layers microstructure in Ta/Cu/SS composite. The authors apply different modern methods (SEM/TEM, EBSD) of microstructure analysis. The technique of explosive welding of Cu/SS and Cu/Ta composites known and presented in the scientific papers. Any commercial clad products are available in the market. The paper is consistent, but information is interesting mostly for industrial practice and for confirmation of known facts.
The combination of Ta/Cu/SS is not the new idea. T. Balzynski talked about such clad plate in the monograph (https://doi.org/10.1007/978-94-011-9751-9). Bouckaert with co-authors analyzed the similar products in monograph (Bouckaert, G.P; Hix, H.B.; Chelius, J. (1974) Explosive-bonded tantalum-steel vessels. DECHEMA Monograph). But authors not show, that microstructure of interfacial and vortex regions of Cu/Ta, Cu/SS and Ta/Cu/SS composites already studied before and presented in the articles. For example: doi:10.1016/j.matdes.2004.07.021; doi: 10.1016/j.matdes.2018.05.027; doi: 10.1080/09276440.2020.1736880; doi: 10.1080/10426914.2012.736665; doi: 10.1109/IFOST.2014.6991157; doi;10.1016/j.matdes.2019.108348
Therefore, the reasons of this research and review of published articles must be improved and related with object of this work and published research results. Authors must explain more detail the main reasons of this study and novelty of presented experimental results, compare them with published information, extend the analysis and comments of main results.
The chapters 4.2 and 4.3 present the information, which is actual for introduction or literature overview chapter. Authors explain but not discuss about achieved microstructural analysis results. The figure 1 is picture of typical for explosive welding technology. There is no big reasons to put such picture in the article. Authors must to revise and improve the summary and general conclusions taking into account main findings. Sections 4.2 and 4.3 have been reorganized and the indicated references were included.
Indeed, some details of the microstructure development in Ta/SS or Ta/Cu/SS clad exist in the literature, e.g. the issue related to the formation of melted zones. However, the evolution of microstructure due to severe plastic deformation and recovery/recrystallization process is barely described in the literature.
Therefore in this paper, we describe in detail the microstructure of both interfaces (Ta/Cu and Cu/SS) using SEM equipped with EBSD facility as well as TEM. Despite the macro/microstructure of Ta/Cu and Cu/SS melted regions were analysed in several papers, there is no information on the strain-hardened layers and their distribution near the interface. In particular, the effect of high temperature on the severely deformed layers and the influence of high strain rate deformation on mechanical twinning in pure copper near the Ta and SS interfaces are quite novel.
However, taking Reviewer suggestions into consideration some parts of the text were revised and the literature modified.

Reviewer 4 Report
Dear Authors,
First of all, CONGRATULATIONS on your work, which is very well conducted and explained. There are no comments regarding formatting and spelling concerns. Regarding technical and scientific concerns, I have some suggestion to do, in order to improve the overall quality of the paper, which is very high even in the current state. Thus, please pay attention to the following comments:
- You are providing some reasons to choose copper to make the intermediate layer. However, I think the reasons could be better dissected. Could you improve the explanation?
- In the Experimental, could you provide the chemical composition of the alloys used in this work? What kind of SS was used? This is important to understand the microstructures obtained further. Moreover, the mechanical properties of the alloys used in the work should be also provided.
- You pointed out 35º as impact angle between plates before explosion. Could you provide information about the selection of this angle?
- You have used an intermediate layer thicker than the flyer plate. Could you provide information about the reasons behind this selection?
- Could you provide information about the microhardness parameters used in the evaluation of the hardness (load, dwell time, hardness type)?
- No information is provided in the Experimental about bending tests, namely the parameters and test configuration. Moreover, in the Results, just one paragraph is devoted to this issue. In the Discussion, the bent samples appear and assume important role. In the Conclusions, the issue is forgotten. Could you homogeneize the information about this subject in all sections?
- You refer "melted zones". Explosion welding is a solid state welding process. Could you explain how the temperature generated through the process was able to melt these zones?
- In the Summary, you refer "crack formation". However, this phenomenon is not dissected before in the paper. How do you explain the cracks occurrence? How can this affect the joint strength?
The quality of the figures is excellent, being in the right proportion throughout the paper. The text is clear and very well written. Thus, it was a pleasure to analyse your paper.
Hope these comments can help you improve the paper overall quality.
Kind regards,
Reviewer
Author Response
Reviewer 4
1. You are providing some reasons to choose copper to make the intermediate layer. However, I think the reasons could be better dissected. Could you improve the explanation?
It has been considered. 2. In the Experimental, could you provide the chemical composition of the alloys used in this work? What kind of SS was used? This is important to understand the microstructures obtained further. Moreover, the mechanical properties of the alloys used in the work should be also provided.
We used a stainless steel of 304L grade. The chemical composition of the metals that were used have been provided in Table 1. 3. You pointed out 35º as impact angle between plates before explosion. Could you provide information about the selection of this angle?
This results from the ‘geometry’ of the shock wave propagation and the dimensions of the plates. The ignition point in the first step of EXW and the situation of the Cu/SS clad used to cover the Ta plate in the second step lead to the conclusion that the directions of wave propagation are inclined at ~35o.
4. You have used an intermediate layer thicker than the flyer plate. Could you provide information about the reasons behind this selection?
For this particular case a relatively thick interlayer was used. This was particularly beneficial for analysing deformation processes inside the copper sheet. Large thickness of the intermediate Cu layer practically eliminates the impact of the Ta/Cu interface on the Cu/SS interface. This was also the reason for covering the Cu/SS clad with Ta in the second EXW step. 5. Could you provide information about the microhardness parameters used in the evaluation of the hardness (load, dwell time, hardness type)?
It has been corrected. Vickers microhardness measurements were performed on the longitudinal sections in order to estimate the hardness across the weld interface. The measurements were performed with a load of 50 G with a dwell time of 15 s. 6. No information is provided in the Experimental about bending tests, namely the parameters and test configuration. Moreover, in the Results, just one paragraph is devoted to this issue. In the Discussion, the bent samples appear and assume important role. In the Conclusions, the issue is forgotten. Could you homogeneize the information about this subject in all sections?
A three-point lateral bending test was performed according to EN 13445-2:2014 (E). The test was performed to evaluate the mechanical properties of the joints. The specimens for bending testing with a size of 10 mm (width) x 13 mm (high) x 150 mm (length) were extracted from the central part of the clad in the plane parallel to the detonation direction. The text in the Discussion and Conclusion was modified accordingly. 7. You refer "melted zones". Explosion welding is a solid state welding process. Could you explain how the temperature generated through the process was able to melt these zones?
The melting of both joined components occurs during EXW since the formation of reaction layers composed mostly of amorphous phases is experimentally observed. It is hence considered that amorphous phases of various chemical compositions cannot be formed as a result of diffusion processes in the solid-state.
The direct measurements of temperature during EXW is very difficult (or impossible). However, recent numerical simulations of the temperature field distribution (e.g. Bataev et al., Mater. Design, 2019) showed that the temperature near the collision line significantly exceeds the melting point of joined materials, even under increased pressure. However, despite this fact, most of the Authors describe EXW as a solid-state metal joining process.
The aforementioned calculations also show that a thin layer of the melt exhibits almost uniform thickness and it is extended along the entire interface. In this approach, the large melted regions are formed due to the continued growth of waves behind the collision point, where the molten metal is squeezed from those areas. This is confirmed by our earlier TEM experimental studies, where the formation of extremely thin films of the melt along the entire interface is formed. This indicates that the dominating mechanism for proper bond formation is based on the localized melting of interfacial layers along the entire interface and it is probably most crucial for ‘proper’ bond. 8. In the Summary, you refer "crack formation". However, this phenomenon is not dissected before in the paper. How do you explain the cracks occurrence? How can this affect the joint strength?
In given metal composition the cracks occur inside the solidified melt regions but they were observed only occasionally.
The cracks formation inside the solidified melt zone is connected with shrinkage of the melt during solidification. Since the chemical composition inside solidified melt zone changes smoothly and the dominating volumes are occupied by non-equilibrium phases (equilibrium phases occurs only accidentally) there is very difficult to state without any doubt which phase (of given chemical composition) favours the cracks nucleation and propagation. The crack occurrence is rather connected with the presence of the amorphous/ultra-fine-grained structure of solidified melt than the crystalline one of the given chemical composition.
The macro-cracks propagate inside the solidified melt zone almost perpendicular to melted zone/parent sheet interface. They usually coexist together with non-regular network of micro- cracks. However, it is important to note that the crack ‘density’ in the Ta/Cu cladding system is lower as compared with other metal compositions.

Round 2
Reviewer 3 Report
This version of article is better. Authors corrected the text according to the all reviewers comments.